# Analysis of Effective Three-Level Neutral Point Clamped Converter System for the Bipolar LVDC Distribution

**Ju-Yong Kim [1], Ho-Sung Kim [2], Ju-Won Baek [2] and Dong-Keun Jeong [2,\*]**

[1] Smart Power Distribution Lab, Power Distribution ICT Group, KEPCO, Daejeon 34056, Korea; juyong.kim@kepco.co.kr

[2] Power Conversion Research Center, Smart Grid Research Division, KERI, Changwon 51543, Korea; khsgt@keri.re.kr (H.-S.K.); jwbaek@keri.re.kr (J.-W.B.)

\* Correspondence: jeongdk0731@gmail.com; Tel.: +82-55-280-1411

**Abstract:** Low-voltage direct current (LVDC) distribution has attracted attention due to increased DC loads, the popularization of electric vehicles, energy storage systems (ESS), and renewable energy sources such as photovoltaic (PV). This paper studies a $\pm750$ V bipolar DC distribution system and applies a 3-level neutral-point clamped (NPC) AC/DC converter for LVDC distribution. However, the 3-level NPC converter is fundamental in the neutral-point (NP) imbalance problem. This paper discusses the NP balance control method using zero-sequence voltage among various solutions to solve NP imbalance. However, since the zero-sequence voltage for NP balance control is limited, the NP voltage cannot be controlled to be balanced when extreme load differences occur. To maintain microgrid stability with bipolar LVDC distribution, it is necessary to control the NP voltage balance, even in an imbalance of extreme load. In addition, due to the bipolar LVDC distribution, the pole where a short-circuit condition occurs limits the short current until the circuit breaker operates, and a pole without a short-circuit condition must supply a stable voltage. Since the conventional 3-level NPC AC/DC converter alone cannot satisfy both functions, an additional DC/DC converter is proposed, analyzed, and verified. This paper is about a 3-level NPC AC/DC converter system for LVDC distribution. It can be used for the imbalance and short-circuit condition in bipolar LVDC distribution through the prototype of the 300 kW 3-level NPC AC/DC converter system and experimented and verified in various conditions.

**Keywords:** bipolar LVDC distribution; 3-level NPC AC/DC converter; zero-sequence voltage; DC/DC converter

---

## 1. Introduction

In recent years, due to resource depletion and environmental problems such as pollution and climate change, renewable energy sources such as photovoltaic (PV) cells and wind energy have been supplied to replace fossil fuels. Along with the increase in renewable energy sources, increasingly, energy storage systems (ESS) and DC loads have led to an interest in the low-voltage direct current (LVDC) distribution [1–3]. Compared with AC distribution, LVDC distribution has several advantages [4–6]. LVDC distribution is easy to connect to renewable energy sources and ESS, which are DC sources. Due to the increased DC load, LVDC distribution can increase energy efficiency by eliminating unnecessary power conversion and minimizing power conversion [7,8]. In addition, by reducing the power conversion process, it is possible to make the equipment lighter and smaller and to improve the reliability of LVDC distribution. Since DC does not have a frequency, there is no reactance component and no reactive power. Therefore, there is no skin effect during distribution.

Besides that, LVDC distribution has advantages such as high transmission capacity, low transmission loss, low environmental impact, and low investment costs [9–11].

Previous studies on the LVDC distribution system have validated its performance and impact in industrial and commercial applications [12,13]. A variety of DC distribution schemes have been proposed [14–16]. The existing AC distribution network consists of 3-phase AC power wires; so, if LVDC distribution is configured as bipolar, it can use a previously installed AC distribution network as a DC distribution network without additional installation [17,18]. There are many economic benefits from the supplier side. The bipolar $\pm 750$ $V_{dc}$ distribution system has been reported to have the advantages of high power dissipation, low conduction losses, and good user safety [19]. In this paper, Figure 1 shows the LVDC distribution system with a bipolar $\pm 750$ $V_{dc}$ output voltage considering the dielectric strength of the previously installed wires.

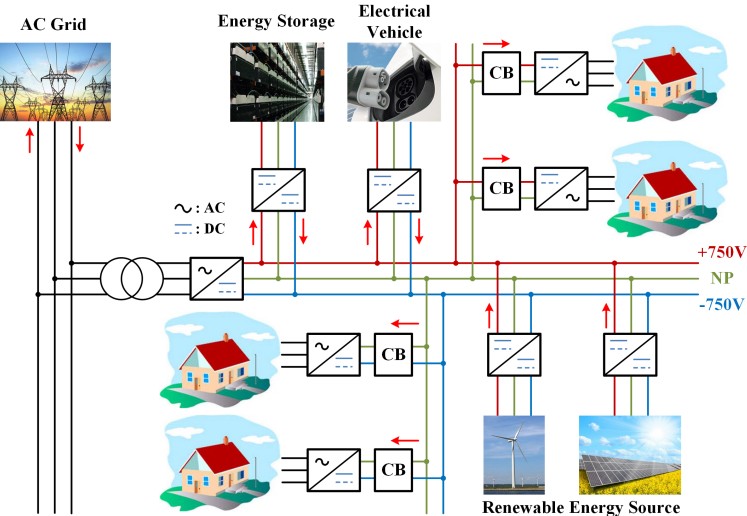

**Figure 1.** Structure of a bipolar low-voltage direct current (LVDC) distribution.

The proposed LVDC distribution requires the bipolar AC/DC converter due to the bipolar power distribution. Several converters have been studied to find a two-line converter for the bipolar power distribution [20–23]. The 12-pulse thyristor converter is suitable for bipolar output converters [20,21]. However, it has a low power factor (PF). Additionally, two separated thyristor converters with a three-winding transformer are required for bipolar power transmission. Although the 2-level AC/DC converter has a higher PF and total harmonic distortion (THD) than the 12-pulse thyristor converter, a three-winding transformer is also needed to apply a bipolar LVDC distribution system [22,23]. Compared with these weaknesses, since it has lower harmonic distortion on the AC-side, the 3-level AC/DC converter has better power quality without additional transformers [24].

Among the various 3-level AC/DC converters, the 3-level neutral-point clamped (NPC) converter offers high reliability, safety, and a low-rated voltage [25–27]. Phase voltage balance can be achieved by an appropriate control strategy. Therefore, the 3-level NPC converter is applied to the AC/DC converter for LVDC distribution.

Three kinds of problems must be overcome in order to utilize the typical 3-level NPC AC/DC converter as two independent sources of the bipolar LVDC distribution system. First, the voltage balance between positive and negative poles is required in the bipolar power distribution system. A 3-level NPC AC/DC converter has an inherent neutral point (NP) voltage balancing problem. It limits the performance of the bipolar $\pm 750$ $V_{dc}$ distribution system. Since customers are separately connected to the positive and negative poles, as shown in Figure 1, this can lead to asymmetrical load conditions between positive and negative poles. Therefore, NP voltage balancing control is required for bipolar $\pm 750$ $V_{dc}$ distribution systems.

In order to solve the DC-link voltage imbalance caused by asymmetrical loads, various methods have been introduced to balance the NP voltage. A method to prevent the imbalanced DC-link voltage is to use an external converter to compensate for the DC component of NP current [28,29]. In this method, the DC-link voltage difference between the positive and negative poles is controlled by a compensator that specifies the amount of current that the external converter must inject at NP. The main disadvantage of this method is that it requires additional power hardware and therefore increases system cost and complexity. The strategy of calculating the zero-sequence voltage including the NP balancing factor can simply compensate for the NP fluctuation under asymmetrical load conditions [30–33]. This method provides an economically feasible and technically sophisticated solution. However, in general, previous research has focused on compensating the NP current generated only in the switching state when using the entire DC-link. In the case of using a bipolar load, there is no study into the required zero-sequence voltage due to the different load between the positive pole and the negative pole. Therefore, this paper analyses the required zero-sequence voltage according to the NP current generated by both the load difference and the switching state. However, in the case of an extremely asymmetrical load, the method of using the zero-sequence voltage is limited. Therefore, an additional circuit should be added for the NP voltage balance since it cannot be adapted to the bipolar $\pm 750$ V$_{dc}$ distribution system.

Secondly, since the ESS or a PV system is also separately integrated into the positive and negative pole, each of the positive and negative poles must individually allow bi-directional operation in the bipolar power distribution system. However, the conventional 3-level NPC AC/DC converter can be operated in both positive and negative directions, but cannot be operated individually.

Finally, the LVDC distribution system requires protection against a short-circuit condition. Since the customer is separately connected to the positive and negative pole, if a short-circuit condition occurs in either the positive or negative pole, a pole which does not feature in the short-circuit condition must normally distribute the power. If the short-circuit condition occurs, the circuit breaker where the short-circuit condition occurs must operate immediately. The short-circuit current limits until the circuit breaker trips to prevent the AC/DC converter from faulting and prevent overcurrent from flowing into the short circuit. In addition, when the short-circuit condition is resolved, it should automatically return to normal operation. However, the conventional 3-level NPC AC/DC converter cannot operate because a large NP current occurs when a pole is short-circuited.

Applying a 3-level NPC AC/DC converter to a bipolar LVDC distribution system should solve these problems. In the paper, the required zero-sequence voltage, including the imbalance of the NP voltage due to the switching operation as well as the imbalance of the NP voltage due to the imbalance of the load applied to the individual poles, is proposed and analyzed. In addition, the limit of zero-sequence voltage is analyzed in bipolar LVDC distribution. To solve this problem, this paper proposes an additional DC/DC converter circuit and a control algorithm that can satisfy the imbalance problem and other conditions to use a 3-level NPC AC/DC converter in a bipolar LVDC distribution. The proposed additional circuit improves NP voltage balancing control for extremely asymmetrical loads. It also enables the individual bi-directional operation of each pole and maintains the bipolar distribution system in short-circuit conditions. The performance of the proposed system is analyzed and experimentally verified using a 300 kVA prototype 3-level NPC AC/DC converter system.

## 2. Three-Level Neutral Point Clamped Converter for LVDC Distribution

Figure 2 is a 3-level NPC AC/DC converter circuit. It consists of four switching devices and two clamping diodes in each leg. The DC-bus voltage in the 3-level NPC AC/DC converter is split into 3-levels by two series-connected bulk capacitors, $C_{npc.p}$ and $C_{npc.n}$. The middle point of the two capacitors NP can be defined as the neutral point. Furthermore, the positive and negative pole voltage were defined as $v_{npc.p}$ and $v_{npc.n}$.

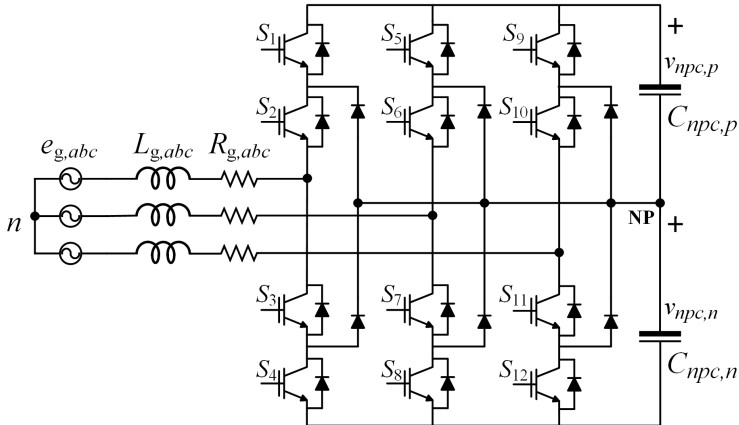

**Figure 2.** The circuit configuration of 3-level neutral-point clamped (NPC) AC/DC converter.

## 2.1. Neutral Point Imbalance

This paper applies the space vector pulse width modulation (SVPWM) to the NPC converter. The NPC converter has three switching states for each leg. Therefore, considering all three phases, the NPC converter can have 27 switching states. The sector can be divided into categories *A* to *F* depending on each switching state as shown in Figure 3a. Switching states can also be classified into three vectors, large vector (LV), medium vector (MV), and small vector (SV), depending on the combination of output phase voltages. There are two types of small vectors: the upper voltage vector (USV) and the lower voltage vector (LSV). The combination of these vectors determines the output voltage of the converter, and the NP current flows differently. LV has no effect on the NP current, and MV does not affect the NP current when the load of each pole is the same. However, SV affects the NP current. Figure 3b describes the current flow of (*OPP*) and (*NOO*) among the SVs according to the switching of the 3-level NPC AC/DC converter. The current flows through the upper capacitor of the DC-link under the switching state of (*OPP*) and flows through the lower capacitor of DC-link under the switching state of (*NOO*). In addition, since it is a converter for LVDC distribution, the NP imbalance caused by the load applied to each pole must also be considered.

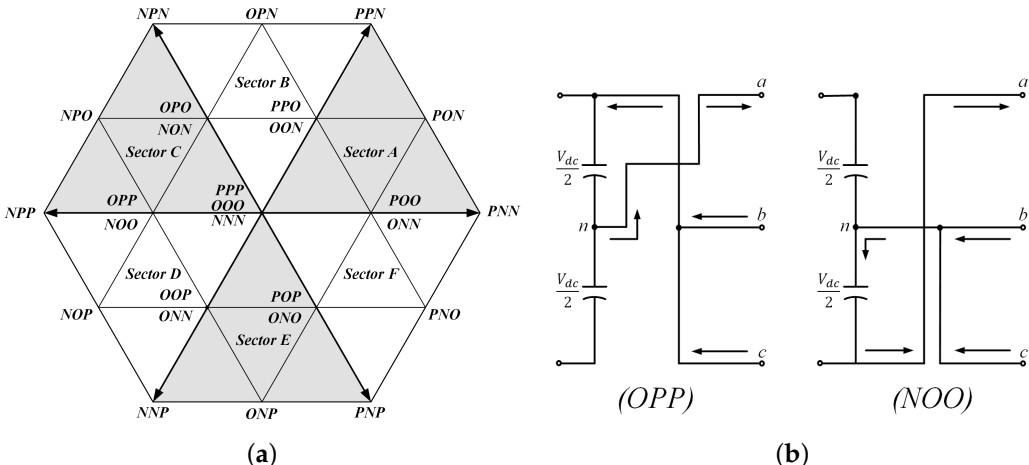

(**a**)

(**b**)

**Figure 3.** Switching Vector and Sector of the 3-level AC/DC converter: (**a**) Space vector pulse width modulation (SVPWM); (**b**) Small Vector *OPP* and *NOO*.

## 2.2. Zero-Sequence Voltage for Neutral Point Potential Balancing

In order to prevent the imbalanced NP voltage, the method injected zero-sequence voltage can be used. The zero-sequence voltage injection method does not affect the AC phase voltage or current in the 3-phase reference. It is also possible to compensate for the effect of the increasing NP voltage

difference through the same utilization of both capacitors. To obtain the duration time of the required SV for NP voltage balancing, a complex calculation process is required. However, the method of injecting the zero-sequence voltage into the reference voltage can easily control the redundant SV without a complicated calculation process. The zero-sequence voltage is a DC component.

Figure 4 shows the vector sequence by injecting the zero-sequence voltage at the 3-phase reference voltage. Even when a zero-sequence voltage is injected, the duration times of LV ($NPP$) and MV ($NPO$) do not change. It affects only the duration time of SV ($OPP$) and ($NOO$), which affects the imbalance of the NP voltage. Therefore, adjusting only the SV can easily balance the NP voltage. The fact that the zero-sequence voltage is added as a positive sign to the reference voltage as shown in Figure 4. It can be expected that the duration time of the SV ($OPP$) in the switching state is less than the duration time of SV ($NOO$), or the load of the positive pole is larger than that of the negative pole.

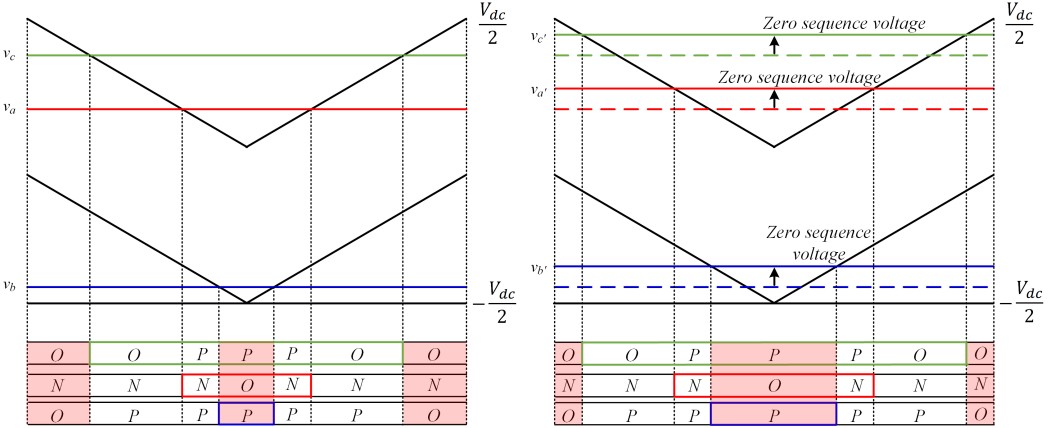

**Figure 4.** The duration time of the small vector ($OPP$) and ($NOO$) when the zero-sequence voltage is added.

However, the zero-sequence voltage cannot be an unconditional solution to the imbalance of DC-link voltage. Figure 5 describes the applicable range and limit of the zero-sequence voltage. The range of the zero-sequence voltage should be limited because the reference voltage must not exceed the bounds of the carrier voltage when injecting the zero-sequence voltage to it, as shown in Figure 5. The minimum and maximum range of the zero-sequence voltage is as follows.

$$-\frac{V_{dc}}{2} - v_{abc,min} \leq v_{zero-sequence} \leq \frac{V_{dc}}{2} - v_{abc,max} \tag{1}$$

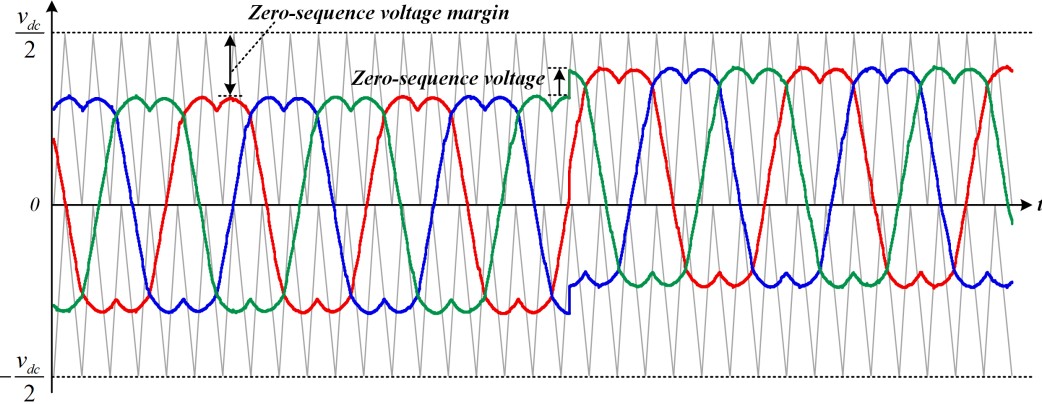

**Figure 5.** Applicable range and limit of zero-sequence voltage.

Therefore, when the loads of positive and negative poles are extremely asymmetry, since the injected zero-sequence voltage method has limit, an additional DC/DC converter is needed to solve the imbalance of DC-link voltage.

*2.3. Mathematical Analysis of Zero-Sequence Voltage according to NP Current*

The control of 3-level NPC AC/DC converters has been studied for the balancing of the NP voltage. However, in general, research has been focused on compensating the NP current generated only in the switching state when using the entire DC-link. In the case of using a bipolar load, there is no study into the required zero-sequence voltage due to the different loads between the positive pole and the negative pole. Therefore, we attempt to derive the required zero-sequence voltage according to the NP current $i_{np}$ generated by both the load difference and the switching state.

The reference voltage of the 3-phase positive sequence, in which the zero-sequence voltage $v_0$ is injected as the reference voltage, can be expressed as

$$
\begin{aligned}
v_a &= M\cos(\theta) + v_0 \\
v_b &= M\cos(\theta - 2\pi/3) + v_0 \\
v_c &= M\cos(\theta + 2\pi/3) + v_0
\end{aligned}
\tag{2}
$$

where $M$ expresses the required amplitude, and $\theta$ contains the phase-angle with frequency. $v_a$, $v_b$, and $v_c$ have the same form but have a phase difference with respect to each other by $2\pi/3$.

$$
\begin{aligned}
i_a &= I_m\cos[\theta - \psi] \\
i_b &= I_m\cos[\theta - \frac{2\pi}{3} - \psi] \\
i_c &= I_m\cos[\theta + \frac{2\pi}{3} - \psi]
\end{aligned}
\tag{3}
$$

where $\psi$ is the phase delay of the AC-side current components with respect to their corresponding AC-side terminal voltages. It can be derived that the DC component of NP current $i_{np}$ can be controlled by the zero-sequence voltage $v_0$ [33].

$$
i_{np,0} = -\frac{6 \times I_m \times \cos(\psi)}{\pi} \times v_0
\tag{4}
$$

The NP current due to the switching vector does not need to consider load angle $\delta$. However, in order to obtain the NP current due to the load difference between the positive pole and the negative pole, a load angle $\delta$ must be considered. Considering $\delta$, Equation (4) can be rewritten as follows.

$$
i_{np,0} = -\frac{6 \times I_m \times \cos(\psi + \delta)}{\pi} \times v_0
\tag{5}
$$

Figure 6 describes the phase diagram according to the load angle and power factor angle. When PF is 1, $\psi$ is zero as shown in Figure 6. Assuming that the 3-level NPC AC/DC converter is fully controlled to have a PF of 1, $\delta$ and $I_m$ can be substituted by other factors. Assuming a considered purely inductive reactance between the two sources, $tan\delta$ can be expressed as

$$
tan\delta = \frac{\omega \times L \times I}{E} = \frac{2\pi \times f \times L \times \frac{P}{V_{rms}}}{V_{rms}}
\tag{6}
$$

We can obtain $\delta$ by taking the tangent inverse from Equation (6).

$$
\delta = \tan^{-1}\left(\frac{2\pi \times f \times L \times \frac{P}{V_{rms}}}{V_{rms}}\right)
\tag{7}
$$

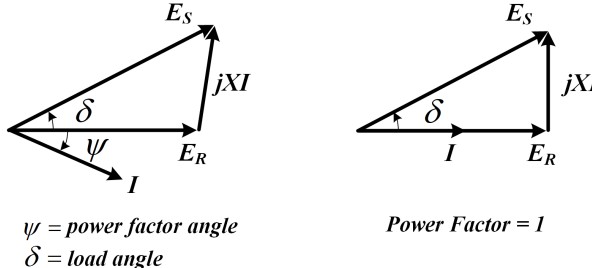

**Figure 6.** Phase diagram according to load angle and power factor angle.

Assuming no losses such as switching losses or conduction losses, the input and output powers are the same. It can be expressed as

$$V_{rms} \times I_{rms} = \frac{V_{dc}}{2}(i_{dc,p} + i_{dc,n})$$

(8)

where $i_{dc.p}$ is the output current of the positive pole, and $i_{dc.n}$ is the output current of the negative pole. Considering $I_{rms}$, Equation (8) can be rewritten as follows:

$$I_{rms} = \frac{V_{dc}}{2V_{rms}}(i_{dc,p} + i_{dc,n})$$

(9)

The relation between $I_m$ and $I_{rms}$ is as follows:

$$I_m = \frac{\sqrt{2} \times I_{rms}}{\sqrt{3}}$$

(10)

where $i_{dc.p}$ is the output current of the positive pole, and $i_{dc.n}$ is the output current of the negative pole. Considering $I_{rms}$, Equation (8), can be rewritten as follows.

$$I_m = \frac{\sqrt{2} \times V_{dc} \times (i_{dc.p} + i_{dc.n})}{2\sqrt{3} \times V_{rms}}$$

(11)

Substituting Equations (7) and (11) into Equation (5), the zero-sequence voltage can be obtained according to the magnitude of the NP current.

$$v_0 = \frac{\sqrt{6} \times \pi \times V_{rms} \times i_{np}}{6 \times V_{dc} \times (i_{dc.p} + i_{dc.n}) \times cos(\frac{2 \times \pi \times f \times L \times P}{V_{rms}^2})}$$

(12)

The NP current generated by the difference between the output currents of positive and negative poles is fatal to the imbalance of the DC-link voltage for LVDC distribution. The zero-sequence voltage required to resolve the imbalance of the DC-link voltage can be obtained from Equation (12) depending on the NP current. Figure 7 shows the required zero-sequence voltage according to the difference between $i_{dc.p}$ and $i_{dc.n}$.

When a constant current flows through $i_{dc.p}$ or $i_{dc.n}$, the required zero-sequence voltage can be displayed depending on the current of the other pole. Figure 8 shows the required zero-sequence voltage according to the current of $i_{dc.p}$ when the current of $i_{dc.n}$ flows continuously to the rated current 200 A in 10 A units. Because the applicable boundary of the zero-sequence voltage is 0.346, if the magnitude of the zero-sequence voltage is greater than 0.346, the zero-sequence voltage alone cannot resolve the imbalance of the DC-link voltage. When the current of $i_{dc.n}$ is 20 A, the zero-sequence voltage can resolve the imbalance of the DC-link voltage at $i_{dc.p}$ below 102 A. However, an $i_{dc.p}$ over 102 A cannot resolve the imbalance of the DC-link voltage because it exceeds the applicable boundary of the zero-sequence voltage as shown in Figure 8. When $i_{dc.n}$ is above 40 A, it also shows that the

imbalance of the DC-link voltage can be resolved by injecting a zero-sequence voltage up to a rated 200 A $i_{dc.p}$.

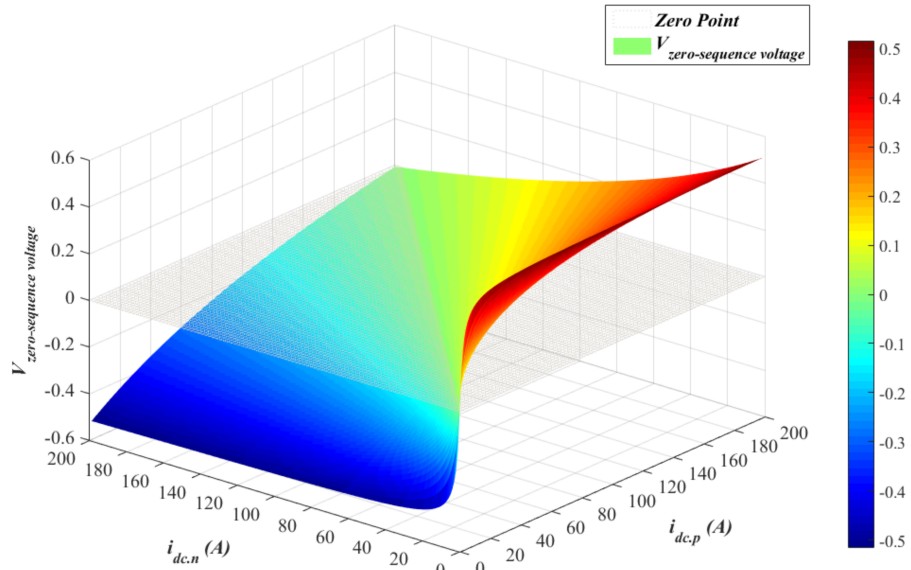

**Figure 7.** The required zero-sequence voltage depending on the difference between the loads of the positive and negative poles.

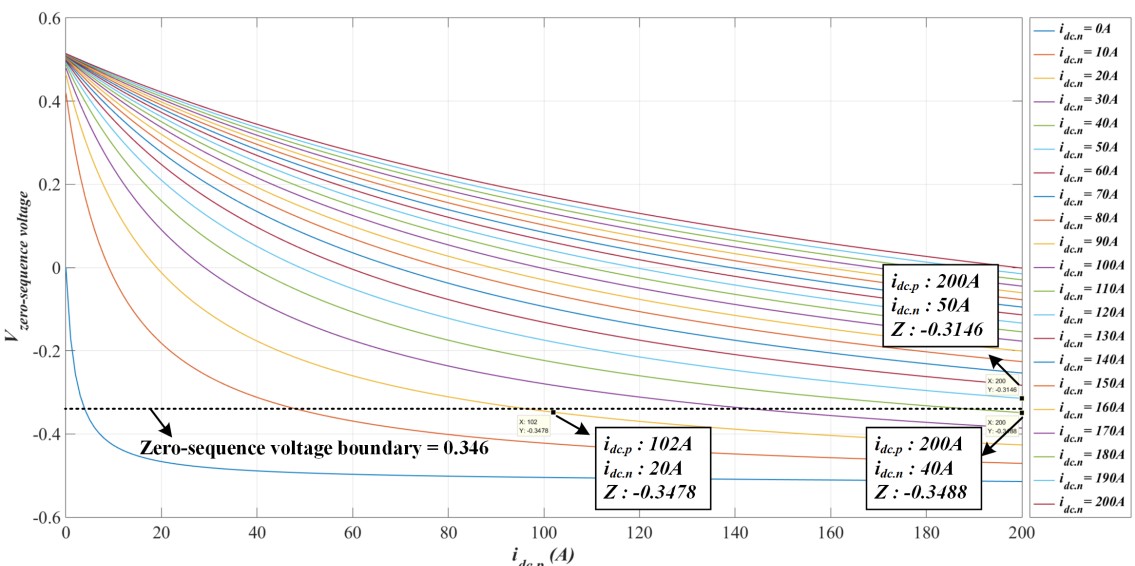

**Figure 8.** The required zero-sequence voltage for the load of the positive pole at constant load of the negative pole.

## 3. Proposed DC/DC Converter for LVDC Distribution

In a bipolar LVDC distribution, the customer is connected individually to a positive pole or a negative pole as shown in Figure 1. In addition, renewable energy sources such as PV or ESS can also be connected individually to the two poles. This means that there are a variety of possible cases. Therefore, the AC/DC converter for bipolar LVDC distribution requires many functions. The essential functions for bipolar LVDC distribution are as follows.

- DC-link voltage balancing under asymmetrical load conditions.
- Independent bi-directional operation of positive and negative poles.
- Preparations for short-circuit condition.

The conventional 3-level NPC AC/DC converter cannot satisfy the above conditions for bipolar LVDC distribution. Since the injected zero-sequence voltage method has limited under extremely asymmetrical load conditions, the imbalance of the DC-link voltage occurs. In addition, if the conventional 3-level NPC AC/DC converter operates in the independent bi-directional condition of positive and negative poles, excessive NP current will be generated as with extremely asymmetrical load conditions. Since the injected zero-sequence voltage method is limited, the conventional 3-level NPC AC/DC converter cannot perform the independent bi-directional operation of positive and negative poles. Since excessive NP current also occurs in the short-circuit condition, the conventional 3-level NPC AC/DC converter cannot control the positive and negative poles. Therefore, this paper proposes the additional DC/DC converter. Figure 9 shows a 3-level NPC AC/DC converter system including the proposed DC/DC converter. The DC/DC converter connects in the series with a 3-level NPC AC/DC converter. It consists of four switches $S_{13}$, $S_{14}$, $S_{15}$, and $S_{16}$, and has three inductors $L_p$, $L_{np}$, and $L_n$.

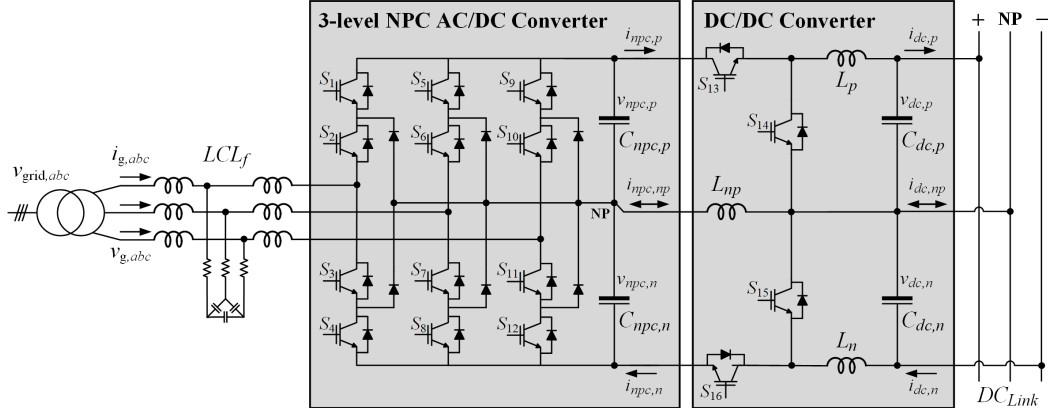

**Figure 9.** The proposed 3-level NPC AC/DC converter system for LVDC distribution.

### 3.1. Advantage of the Proposed DC/DC Converter

Figure 10 shows that the AC/DC converter for the bipolar LVDC distribution can be composed of two 2-level AC / DC converters or a 3-level AC/DC converter. A 2-level AC/DC converter requires a three-winding transformer. Two 2-level AC/DC converters do not cause an imbalance problem in the bipolar output voltage. However, two additional DC/DC converters are eventually required to satisfy the above functions for bipolar LVDC distribution, as shown in Figure 10a.

The 3-level NPC AC/DC converter requires six more diodes than two 2-level AC/DC converters, but does not need the three-winding transformer and reduces the required input filter. In addition, although the 3-level NPC AC/DC converters have the inherent problem of NP voltage imbalance, as shown in Figure 10b, one proposed DC/DC converter satisfies the essential functions for bipolar LVDC distribution, which has advantages in terms of price and equipment volume.

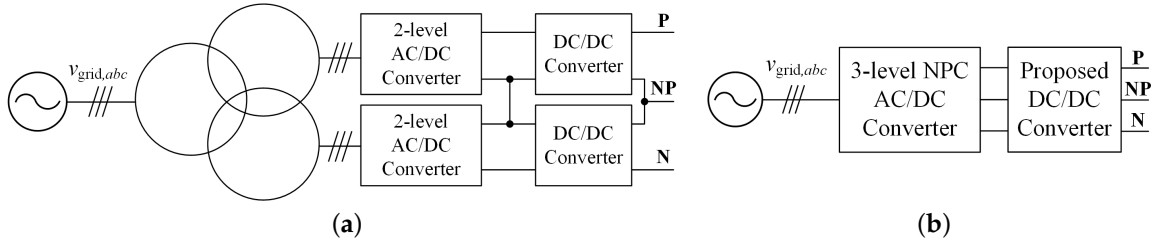

(**a**)　　　　　　　　　　　　　　　　　　　　　　　　　(**b**)

**Figure 10.** Composed of AC/DC converter system for the bipolar LVDC distribution: (**a**) 2-level AC/DC converter; (**b**) 3-level AC/DC converter.

### 3.2. DC-Link Voltage Balancing Operation

Since the injected zero-sequence voltage cannot resolve the imbalance of the DC-link voltage at extreme loads or bi-directional operation of individual poles, in order to maintain a stable microgrid for the bipolar LVDC distribution, the DC/DC converter needs the balancing function of the DC-link voltage. The switches $S_{14}$ and $S_{15}$ operate for DC-link voltage balancing. Switches $S_{13}$ and $S_{16}$ always remain in the *ON* state during the normal state and the balancing operation of the DC-link voltage.

Figure 11 describes the balancing operation of the DC/DC converter. The load condition is that the load of the positive pole $z_p$ is larger than that of the negative pole $z_n$. If the voltage difference increases beyond a certain level $v_{diff,ref}$ under extreme load conditions, the switch $S_{14}$ or $S_{15}$ will operate for the balancing of DC-link voltage. Switch $S_{15}$ operates when $z_p$ is larger than $z_n$. In the opposite case, switch $S_{14}$ is activated. Figure 11a describes three operating modes of the balancing operation during one switching cycle. In addition, Figure 11b describes the operation waveforms in a steady-state.

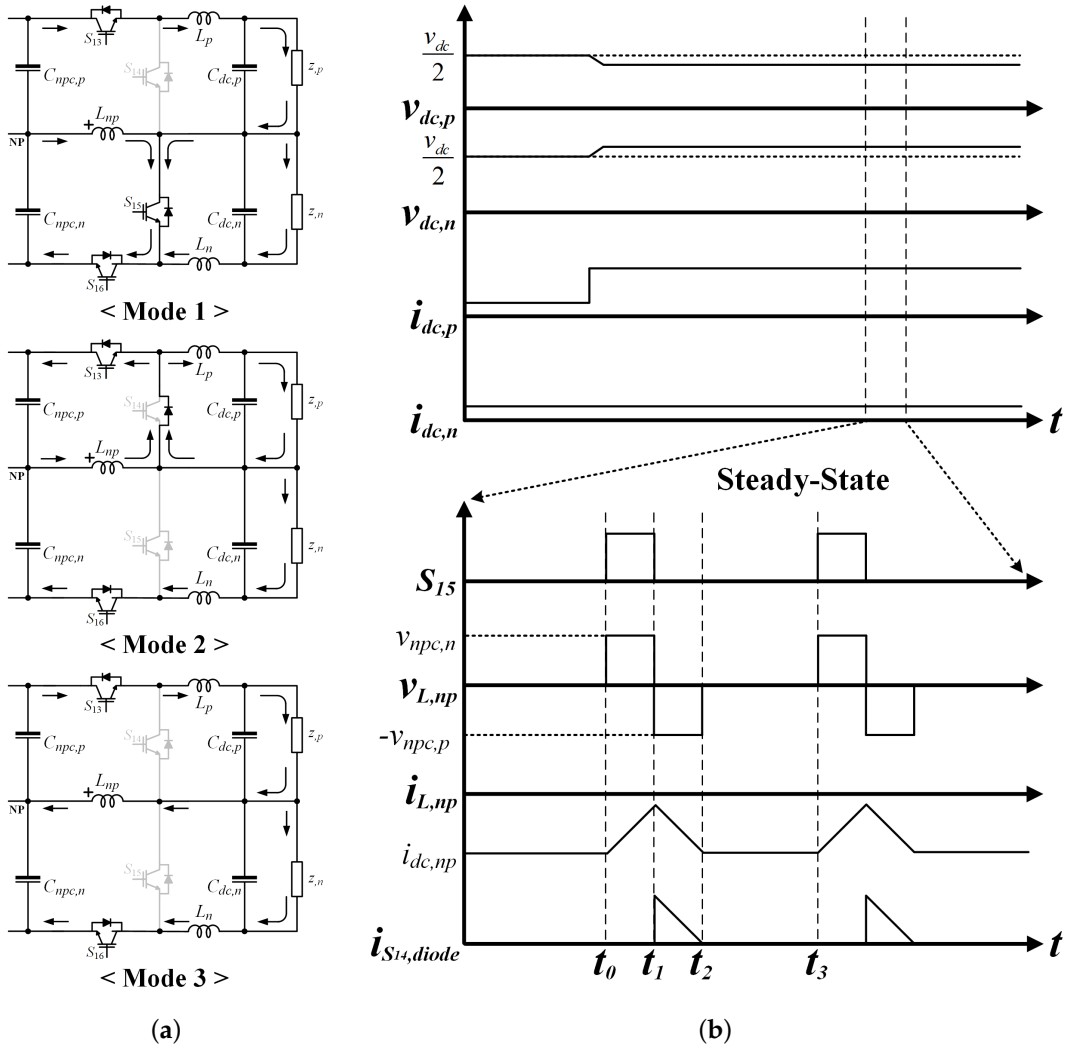

**Figure 11.** Balancing operation of the DC/DC converter at $z_p \gg z_n$ load condition: (**a**) Mode; (**b**) Waveforms.

**Mode 1($t_0$–$t_1$):** When $z_p$ is larger than $z_n$, so that the injected zero-sequence voltage cannot balance the DC-link voltage, a voltage difference occurs under extreme load conditions. If the voltage difference increases by more than $v_{diff,ref}$, the switch $S_{15}$ turns on. The output voltage of the negative pole of the NPC AC/DC converter $v_{npc.n}$ is applied to the inductor $L_{np}$.

**Mode 2($t_1$–$t_2$):** When $S_{15}$ turns off, the current flows through the diode of $S_{14}$ as shown in Figure 11a. The output voltage of the positive pole of the NPC AC/DC converter $-v_{npc,p}$ is applied to $L_{np}$. The power of the capacitor on the negative pole is transferred to the capacitor on the positive pole through $L_{np}$. Therefore, the decreased output voltage of the positive pole due to the large load can be compensated.

**Mode 3($t_2$–$t_3$):** The generated NP current $i_{np,dc}$ due to the load difference between the positive pole and negative pole flows through $L_{np}$. It causes an imbalance of the DC-link voltage.

The inductance $L_{np}$ required for balancing operation can be selected as the desired current ripple $\Delta i_{L_{np}}$ as follows:

$$L_{np} = \frac{v_{npc,n} \times D}{f_{sw} \times \Delta i_{L_{np}}} \tag{13}$$

where $f_{sw}$ is the switching frequency. The balancing operation is used to suppress the NP current. It satisfies the balancing condition of DC-link voltage under the asymmetrical load conditions. In addition, it also allows the independent bi-directional operation of positive and negative poles by suppressing the NP current.

*3.3. Current Limit Operation*

For bipolar LVDC distribution, if a short-circuit condition occurs on the positive pole or a negative pole, the circuit breaker should operate only in the pole where the short-circuit condition occurs. In addition, the pole in which the remaining short-circuit condition does not occur should provide stable power. The conventional 3-level NPC AC/DC converter generates a fault when a short-circuit condition occurs at either the positive pole or a negative pole due to the increasing NP current. Therefore, when a short-circuit condition occurs at either the positive pole or a negative pole, the pole where the short-circuit condition occurs requires the current limit operation by the DC/DC converter. In addition, the pole without any remaining short-circuit condition requires a solution that can reliably supply power. The switches $S_{13}$ and $S_{16}$ operate for apparent short-circuit conditions. Switches $S_{14}$ and $S_{15}$ do not operate during the current limit operation.

Figure 12 describes the current limit operation of the DC/DC converter at the short-circuit condition of the positive pole. Switch $S_{13}$ operates in the case of a positive pole short-circuit condition and Switch $S_{16}$ operates in a negative pole short-circuit condition. When the short-circuit condition occurs at the positive pole, the current limit function of the DC/DC converter is activated by the current of the positive pole over the specified $i_{limit,ref}$. In the transient state, both the switches $S_{13}$ and $S_{16}$ are operated. $S_{13}$ operates for current limit operation, and $S_{16}$ operates to regulate the output voltage of negative pole $v_{dc,n}$. In a steady state, $S_{13}$ continues to operate for current limit operation. Since the $v_{npc,n}$ of the NPC AC/DC converter is the same as $v_{dc,n}$, $S_{16}$ does not operate and remains in an *ON* state. Figure 12a describes three operating modes of the current limit operation during one switching cycle. In addition, Figure 12b describes the operation waveforms in a steady-state.

**Mode 1($t_0$–$t_1$):** $S_{13}$ turns off to limit the current of the positive pole, as shown in Figure 12b. In addition, when $S_{13}$ is turned off, the total DC-link voltage of the NPC AC/DC converter is applied to $S_{13}$. Therefore, $S_{13}$ and $S_{16}$ must have a rated collector-emitter voltage of the switch greater than the DC-link voltage. The $-v_{dc,p}$ of the DC/DC converter is applied to $L_p$. In addition, $v_{dc,n}$ is applied to the $L_n$ and is also applied to the $L_{np}$. Despite the short-circuit condition of the positive pole, only the limited current $i_{limit,ref}$ flows through the positive pole due to the current limit operation of the DC/DC converter. $i_{dc,n}$ required for the load of the negative pole, flows only to the negative pole and the remaining current $i_{dc,np}$ flows to the diode of $S_{14}$. $i_{L,np}$, caused by the short-circuit condition, flows through the diode of $S_{15}$.

**Mode 2($t_1$–$t_2$):** $S_{13}$ turns on. Even though $S_{13}$ is an *ON* state, since $i_{L.np}$ flows through the diode of $S_{14}$, $-v_{dc,p}$ is applied to $L_p$. In addition, $-v_{npc,p}$ is applied to the $L_n$ and $L_{np}$. Therefore, $i_{L.np}$ is decreased and flows through the diode of $S_{14}$.

**Mode 3($t_2$–$t_3$):** $S_{13}$ remains in an *ON* state. The DC current $i_{limit_{ref}}$ limited by the switching operation of $S13$ flows to NP, except the load current $i_{dc.n}$ of the negative pole. $v_{npc,p} + v_{L,n} - v_{dc,p}$ is applied to $L_p$. $v_{L,p} + v_{dc,p} - v_{npc,p}$ is applied to the $L_n$ and $L_{np}$. $v_{npc,p}$ can be assumed to be 750 V.

The inductances $L_n$ required for current limit operation can be chosen as desired current ripple $\Delta i_{L_n}$ as follows

$$L_n = \frac{v_{dc,n} \times (1-D)}{f_{sw} \times \Delta i_{L_n}} \tag{14}$$

$L_p$ is equivalent to $L_n$. If a short-circuit condition occurs, the operation of the DC/DC converter can limit the current of the pole. The output current is controlled by $i_{limit_{ref}}$ at the pole where the short-circuit condition occurs. In addition, the voltage of the pole where the short-circuit condition does not occur is controlled to 750 V to supply power stably.

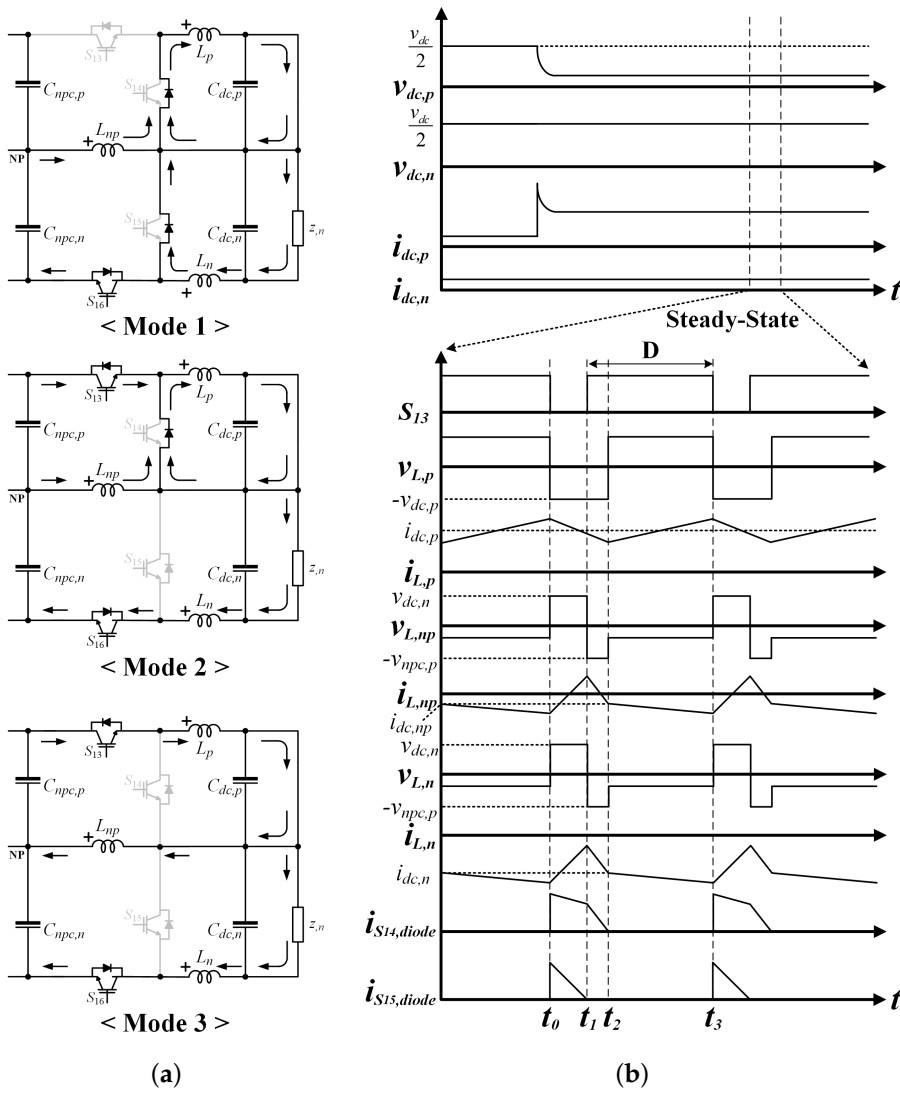

**Figure 12.** Current limit operation at the short-circuit condition of the positive pole: (**a**) Mode; (**b**) Waveforms.

*3.4. Control Algorithm*

Figure 13 describes the control block diagram of the 3-level NPC AC/DC converter system. The 3-level NPC AC/DC converter and the DC/DC converter have individual microcontrollers. In this paper, the designed current and voltage controller are configured in series to control the 3-level NPC AC/DC converter. The DC-link voltage, input current, and a phase locked loop (PLL) are controlled through coordinate conversion by sensing the 3-phase AC grid voltage and current of the system. In addition, a SVPWM is applied as the switching method. The DC-link voltage balance controller calculates the voltage difference between the positive and negative pole. The voltage difference is controlled by the PI controller to 0. From Equation (12), the zero-sequence voltage required to compensate for the NP current due to the difference in load between the positive and negative poles can be obtained. The zero-sequence voltage is calculated by sensing the currents of the positive and negative poles. The feedforward control is implemented by adding the calculated zero-sequence voltage to the PI controller. The final zero-sequence voltage is added to the SVPWM output reference to enable DC-link voltage balance control.

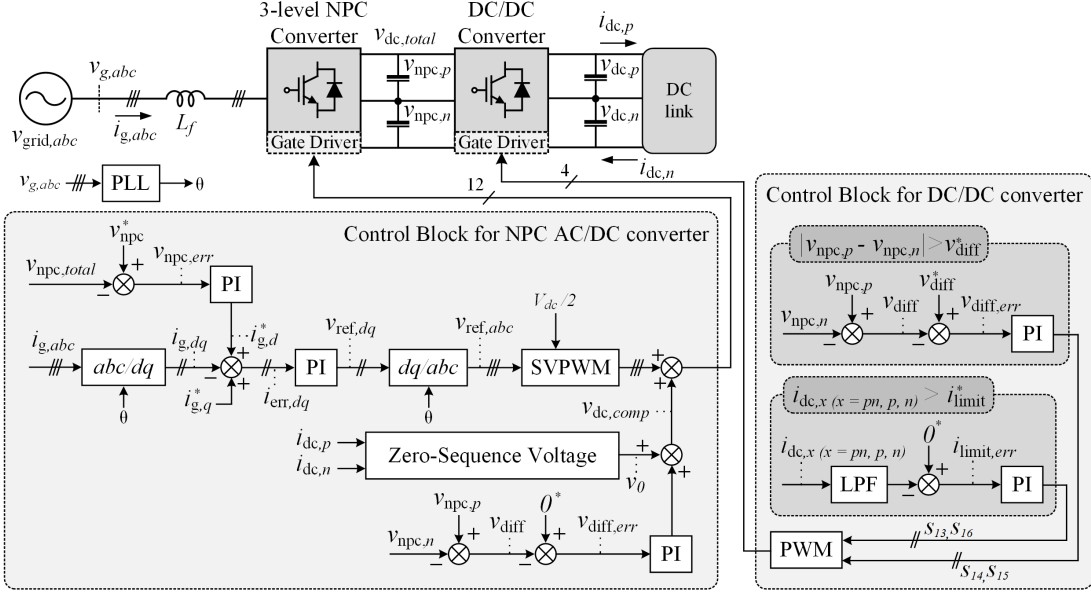

**Figure 13.** Control block diagram of the proposed 3-level NPC AC/DC converter system.

The injected zero-sequence voltage is limited by extreme load difference. When the limit of the zero-sequence voltage is reached, the voltage difference between the positive and negative pole begins to occur. If the voltage difference exceeds a certain level of $v_{diff}$, the DC/DC converter will operate the balancing function. In addition, if the current of the positive or negative pole exceeds a certain level $i_{limit}$, the DC/DC converter will control the short-circuit current to a value of $i_{limit}$.

## 4. Experimental Results

The 3-level NPC AC/DC converter system consists of the 3-level NPC AC/DC converter and the proposed DC/DC converter. Figure 14 describes a 300 kW prototype of the 3-level NPC AC/DC converter system for bipolar LVDC distribution. The power stack of the 3-level NPC AC/DC converter is located on the front side, as shown in Figure 14a, while the *DSP* controller and the power stack of the DC/DC converter are located at the rear, as shown in Figure 14b. The input *LCL* filter the of AC/DC converter and the inductors of DC/DC converter are located under the power stack. Table 1 shows the design specifications for the 3-level NPC AC/DC converter to verify the performance of the proposed 3-level NPC converter system for LVDC distribution. In addition, the design specifications of the proposed DC/DC converter are also shown in Table 2.

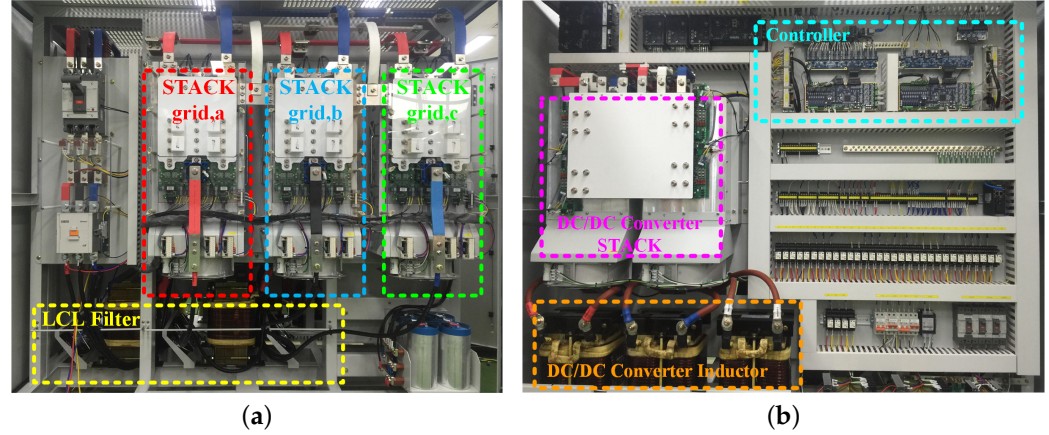

(**a**)                                              (**b**)

**Figure 14.** Prototype 3-level NPC AC/DC converter system for LVDC distribution: (**a**) forward; (**b**) backward.

**Table 1.** Design specifications of the 3-level neutral-point clamped (NPC) AC/DC converter parameters.

| Parameter | Symbol | Value | Unit |
|---|---|---|---|
| Rated output power | $P_{\text{out}}$ | 300 | kW |
| Each pole output voltage | $v_{\text{npc,i}}, i = p, n$ | 750 | V |
| Total output voltage | $v_{\text{npc}}$ | 1500 | V |
| Output current | $i_{\text{npc,i}}, i = p, n$ | 200 | A |
| Input apparent power | $P_{\text{out}}$ | 300 | kVA |
| Input line voltage | $v_{\text{g,i}}, i = a, b, c$ | 600 | Vrms |
| Input vine current | $i_{\text{g,i}}, i = a, b, c$ | 300 | Arms |
| Output capacitors | $C_{\text{npc,i}}, i = p, n$ | 5 | mF |
| Filter inductor | $L_{\text{f}}$ | 300 | uH |
| Filter capacitors | $C_{\text{f}}$ | 33 | uF |
| Switching frequency | $f_{\text{sw,rec}}$ | 6 | kHz |

**Table 2.** Design specifications of the DC/DC converter parameters.

| Parameter | Symbol | Value | Unit |
|---|---|---|---|
| Rated output power | $P_{\text{out}}$ | 300 | kW |
| Each pole output voltage | $v_{\text{dc,i}}, i = p, n$ | 750 | V |
| Total output voltage | $v_{\text{dc}}$ | 1500 | V |
| Each pole Input voltage | $v_{\text{npc,i}}, i = p, n$ | 750 | V |
| Total input voltage | $v_{\text{npc}}$ | 1500 | V |
| Inductor | $L_{\text{i}}, i = p, n, np$ | 800 | uH |
| Output capacitors | $C_{\text{dc,i}}, i = p, n$ | 8 | mF |
| Switching frequency | $f_{\text{sw,rec}}$ | 5 | kHz |

Figure 15 shows the DC-link voltage balancing operation using a zero-sequence voltage. $i_{dc.n}$ flows at 20 A, and the load of the positive pole increases. The zero-sequence voltage injection method is limited by the difference in load. Similar to the zero-sequence voltage calculated at the $i_{dc.p}$ of 102 A from Equation (12), the imbalance of the DC-link voltage can no longer be resolved and a voltage difference occurs as shown in Figure 8. If it is a conventional 3-level NPC AC/DC converter, the imbalance of the output voltage is consistently severe due to the limitation of the injected zero-sequence voltage method, entire system will fault and stop. However, the proposed DC/DC converter constantly controls the voltage difference to 50 V when a voltage difference of more than a certain level occurs.

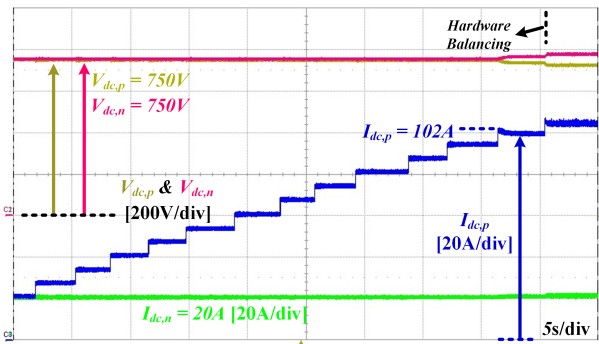

**Figure 15.** Experimental results of the balanced DC-link voltage using zero-sequence voltage and limitation: 102 A $i_{dc,p}$ and 20 A $i_{dc,n}$.

Figure 16 shows the applicable boundary of the zero-sequence voltage. When $i_{dc.n}$ is above 43 A, the zero-sequence voltage calculated from Equation (12) can balance the DC-link voltage irrespective of $i_{dc.p}$. Figure 16a shows that when $i_{dc.n}$ constantly flows at 40 A, the imbalance of the DC-link voltage occurs at an $i_{dc.p}$ of 200 A. However, when $i_{dc.n}$ is greater than 43 A, Figure 16b shows a stable DC-link voltage balancing at $i_{dc.p}$ of 200 A.

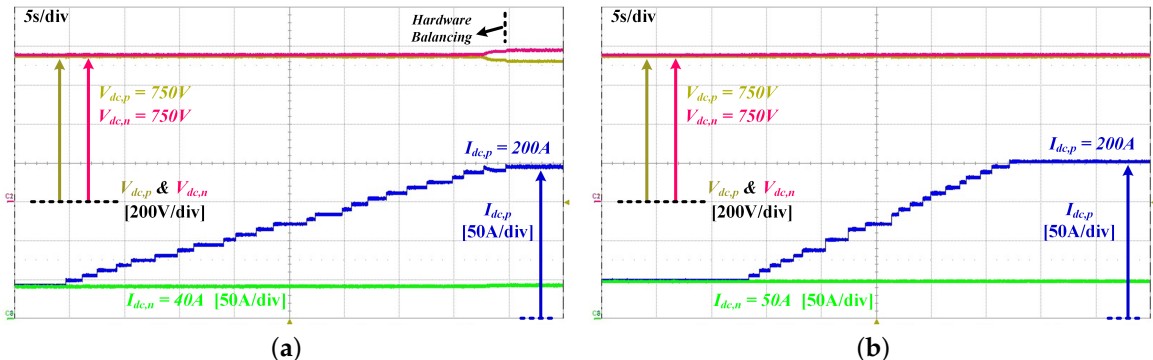

**Figure 16.** Experimental results of the boundary of balanced DC-link voltage using zero-sequence voltage: (**a**) 200 A $i_{dc,p}$ and 40 A $i_{dc,n}$; (**b**) 200 A $i_{dc,p}$ and 50 A $i_{dc,n}$.

Figure 17a shows the experimental waveforms of the balancing operation of the DC/DC converter. Figure 17a verifies the voltage balancing function of the proposed DC/DC converter under extremely heavy asymmetric load conditions. The load on the positive pole is a 100% load and the negative pole is no load. The 3-level NPC AC/DC converter system reduces the voltage difference between the positive and negative pole and maintains the voltage difference of 50 V. In addition, the 3-level NPC AC/DC converter system also regulates the output voltage to 50 V under the extremely heavy asymmetric load condition of the negative pole. The 3-level NPC AC/DC converter is a converter with positive and negative pole outputs. Both poles of the typical 3-level NPC AC/DC converter can be in bi-directional operation in the same direction, although it is not possible to operate the two poles separately bi-directionally. The proposed DC/DC converter solves this problem for LVDC distribution. Figure 17b shows that the power flow of the positive pole is the powering operation and the power flow of the negative pole is the regeneration operation. The 3-level NPC AC/DC converter system prevents the imbalance of the DC-link voltage and maintains the voltage difference at 50 V.

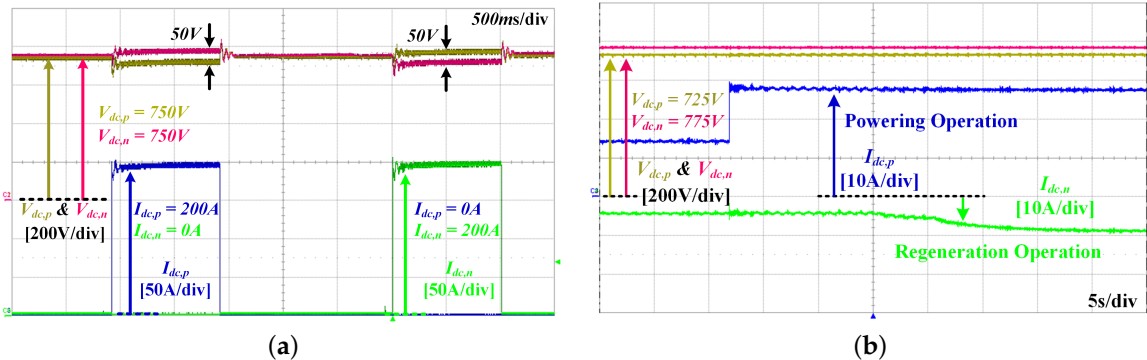

**Figure 17.** Experimental results of the balancing operation of DC/DC converter: (**a**) Extremely heavy asymmetric load conditions; (**b**) Bi-directional unbalancing condition.

In short-circuit condition, the excessive NP current flows to NP. The conventional 3-level NPC AC/DC converters cannot control excessive NP current due to short-circuit condition, entire system will fault and stop. Figure 18 shows the current limit function of the proposed DC/DC converter in case of short-circuit condition. The short-circuit current was tested above 120 A under experimental conditions. Figure 18a,b verifies that the current limit function respectively operates at short-circuit condition in the positive or negative pole. When the short-circuit condition of 160 A occurs in the positive pole, the current of the positive pole is limited to 120 A, and the voltage of the positive pole drops to the proper voltage to limit the 120 A, as shown in Figure 18a. At this time, since the negative pole is not short-circuited, the load connected to the negative pole must be supplied with stable power. Therefore, the imbalanced output voltage caused by the short-circuit condition is controlled stably 750 V at the negative pole. When the short-circuit condition of 135 A occurs in the negative pole, the current of the negative pole is also limited to 120 A, as shown in Figure 18b. Since the negative pole is not short-circuited, the output voltage of the positive pole is regulated by the DC/DC converter. Figure 19 verifies the current limit function under the short-circuit condition of both poles. When the short-circuit condition of 175 A occurs in the positive pole, the current limit function is operated to 120 A. In addition, while the proposed DC/DC converter operates the current limit function at the positive pole, if the short-circuit condition also occurs at the negative pole, the current of the negative pole is also limited as 120 A reference current, as shown in Figure 19.

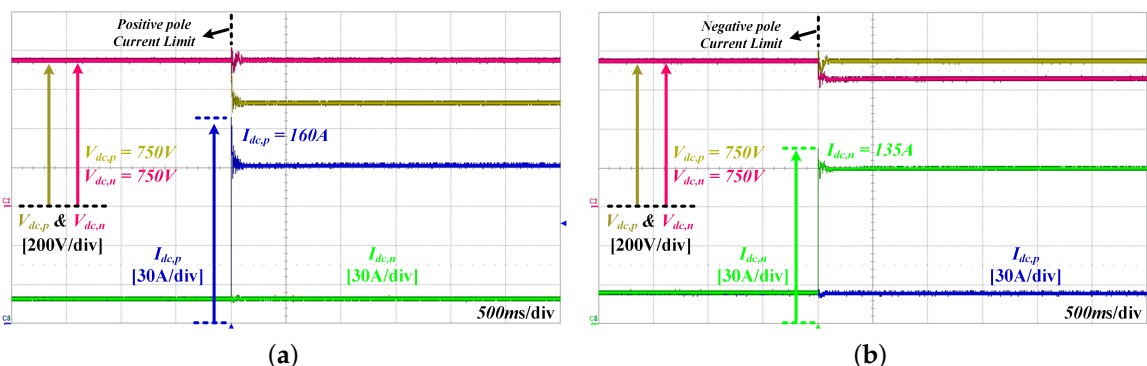

**Figure 18.** Experimental results of current limit operation in short-circuit condition: (**a**) Positive pole; (**b**) Negative pole.

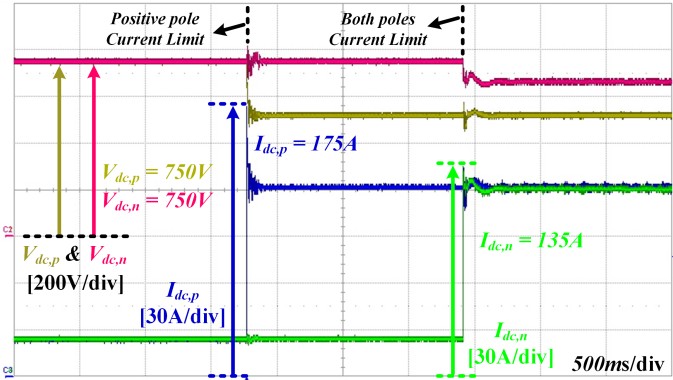

**Figure 19.** Experimental results of current limit operation in short-circuit condition of both poles.

Figure 20 verifies the operation of the proposed 3-level NPC AC/DC converter system under various conditions. Under the slight asymmetric load condition, the injected zero-sequence voltage can regulate the DC-link voltage to equilibrium. However, if the imbalance of the DC-link voltage deviates from the level that can be solved by the zero-sequence voltage, the proposed DC/DC converter reduces the voltage difference between the positive and negative poles to less than 50 V under extremely imbalanced load conditions. When a short-circuit condition occurs in either the positive or negative pole, or both, the proposed DC/DC converter operates to limit the output current as 120 A. The output voltage of the pole without a short-circuit condition is regulated stably at 750 V. If the short-circuit condition disappears due to the circuit breaker operation at the positive pole, the output voltage of the positive pole is normally regulated to 750 V. The proposed 3-level NPC AC/DC converter system operates robustly under various conditions that may occur in the LVDC distribution as shown in Figure 20.

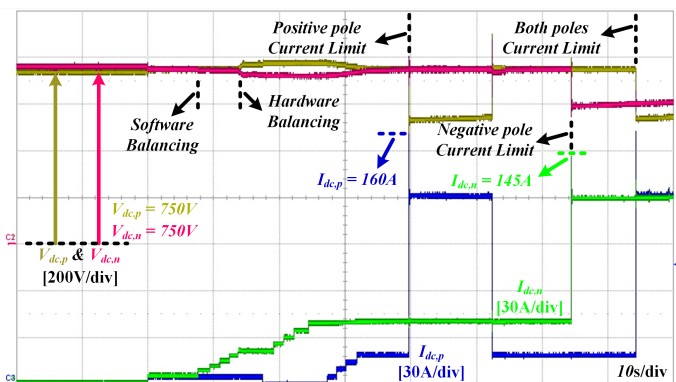

**Figure 20.** Experimental results of the entire function.

## 5. Conclusions

This paper proposes a ±750 V bipolar LVDC distribution system. Among the various AC/DC converters, this paper studies and applies a 3-level NPC AC/DC converter, and due to the inherent NP voltage imbalance, analyzes the NP current generated by the switching state and load difference. In order to solve the imbalance of the NP voltage, this paper discusses the NP balance control method using zero-sequence voltage. The zero-sequence voltage injection method has limitations depending on the difference in load. Since the zero-sequence voltage for NP balance control is limited, the NP voltage cannot be controlled to be balanced when extreme load differences occur. In order to stably maintain the microgrid by using bipolar LVDC distribution, it is necessary to have a function to control the NP voltage balance even in the imbalance of extreme load. The 3-level NPC AC/DC converter is also required for the bi-directional operation of individual poles for bipolar LVDC distribution. In addition, it is also required to limit the current until the circuit breaker trips in the event of a short-circuit

condition. However, a conventional 3-level NPC AC/DC converter does not satisfy these functions. Therefore, in this paper, since the 3-level NPC AC/DC converter alone cannot satisfy these functions, an additional DC/DC converter is proposed and is analyzed among various conditions. The proposed 3-level NPC AC/DC converter system has been investigated and verified under various conditions in the imbalance, individual bidirectional operation, and short-circuit conditions that can occur in bipolar LVDC distribution through the prototype of the 300 kW 3-level NPC AC/DC converter system.

**Author Contributions:** All the authors gave equal contributions in writing and revising the paper.

**Acknowledgments:** This research was supported by the KEPCO Research Institute under the project entitled by Design of analysis model and optimal voltage for MVDC distribution system (R17DA10).

**Conflicts of Interest:** The authors declare no conflict of interest.

## Abbreviations

The following abbreviations are used in this manuscript:

| | |
|---|---|
| ESS | energy storage system |
| LSV | lower voltage vector |
| LV | large vector |
| LVDC | low-voltage direct current |
| MV | medium vector |
| NP | neutral point |
| PF | power factor |
| PLL | phase locked loop |
| PV | photovoltaic |
| SV | small vector |
| SVPWM | space vector pulse width modulation |
| THD | total harmonic distortion |
| USV | upper voltage vector |

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
