# Peer review of "Analysis of Effective Three-Level Neutral Point Clamped Converter System for the Bipolar LVDC Distribution"

_electronics, doi:10.3390/electronics8060691_

Round 1

Reviewer 1 Report

Very interesting paper about Three-level Neutral Point Clamped converter system for the bipolar Low Voltage DC distribution system. Experimental prototype is developed.

Theoretical background is well explained. Figures and diagrams are O.K.

1-      A nomenclature should be provided at the beginning or at the end of the paper to define all variables clearly.

2- The main contribution of the paper shall be more highlighted and emphasized. 

3- It would be great if the drawbacks and gaps of literature are clear and, particularly, how the proposed approach aims at filling these gaps

Author Response

Thanks for addressing English writing Problem. We did a complete proofreading of the paper by MDPI English editing services. Even after extensive English correction by them, we still repeatedly checked our manuscript to provide our paper with English correction as much as possible.

1. A nomenclature should be provided at the beginning or at the end of the paper to define all variables clearly.

Answer

Thanks for the comment. I added a nomenclature at the end of the paper according to your comment. The added nomenclature is as follow:

Abbreviations

The following abbreviations are used in this manuscript

ESS            energy storage system

LSV             lower voltage vector

LV               large vector

LVDC          low-voltage direct current

MV              medium vector

NP               neutral point

PF               power factor

PLL             phase locked loop

PV               photovoltaic

SV               small vector

SVPWM      space vector pulse width modulation

THD            total harmonic distortion

USV            upper voltage vector

2. The main contribution of the paper shall be more highlighted and emphasized.

Answer

Thanks for the comment. In order to emphasize the main contribution, we have added and revised the following sentence on line 94-100, line 141-143 and subsection of updated manuscript.

(line 94-100)

Applying a 3-level NPC AC/DC converter to a bipolar LVDC distribution system should solve these problems. In the paper, the required zero-sequence voltage, including the unbalance of the NP voltage due to the switching operation as well as the unbalance of the NP voltage due to the unbalance of the load applied to the individual poles, is proposed and analyzed. In addition, the limit of zero-sequence voltage is analyzed in bipolar LVDC distribution. To solve this problem, this paper proposes an additional DC/DC converter circuit and a control algorithm that can satisfy the unbalance problem and other conditions to use a 3-level NPC AC/DC converter in a bipolar LVDC distribution.

(line 141-143)

Therefore, when the loads of positive and negative poles are extremely asymmetry, since the injected zero-sequence voltage method has limit, an additional DC/DC converter is needed to solve the unbalance of DC-link voltage.

(line 189-199)

Subsection(3.1. Advantage of the proposed DC/DC converter)

Figure. 10 shows that the AC/DC converter for the bipolar LVDC distribution can be composed of two 2-level AC / DC converters or a 3-level AC/DC converter. A 2-level AC/DC converter requires a three-winding transformer. Two 2-level AC/DC converters do not cause an unbalance problem in the bipolar output voltage. However, two additional DC/DC converters are eventually required to satisfy the above functions for bipolar LVDC distribution, as shown in Figure. 10(a).

The 3-level NPC AC/DC converter requires six more diodes than two 2-level AC/DC converters, but do not needed the three-winding transformer and reduces the required input filter. In addition, although the 3-level NPC AC/DC converters have the inherent problem of NP voltage unbalance, as shown in Figure. 10(b), one proposed DC/DC converter satisfies the essential functions for bipolar LVDC distribution, which has advantages in terms of price and equipment volume.

3. It would be great if the drawbacks and gaps of literature are clear and, particularly, how the proposed approach aims at filling these gaps

Answer

Thanks for the comment. As comment by the reviewer, in order to explain how the proposed approach aims at filling these gaps. we have added the following sentence on lines 177-184 of updated manuscript.

(line 177-184)

Since the injected zero-sequence voltage method has limited under extremely asymmetrical load conditions, the unbalance of the DC-link voltage occurs. In addition, if the conventional 3-level NPC AC/DC converter operate in the independent bi-directional condition of positive and negative poles, excessive NP current will be generated as with extremely asymmetrical load conditions. Since the injected zero-sequence voltage method has limited, the conventional 3-level NPC AC/DC converter can not perform the independent bi-directional operation of positive and negative poles. Since excessive NP current also occurs in the short-circuit condition, the conventional 3-level NPC AC/DC converter can not control the positive and negative poles.

Reviewer 2 Report

This paper presents a Analysis of Effective 3-level Neutral Point Clamped converter system for the bipolar LVDC distribution. The unbalance and short circuit problem in LVDC distribution system has been studies and analyzed. The DC/DC Converter has been proposed for balance conditions and controlled using zero-sequence voltage through load difference. To enhance and verify the proposed method, prototype of the 300Kw 3-level NPC AC/DC converter system has been tested during various load conditions. The paper sounds interesting and has a clearer structure but some questions need to be answered and notes should be considered: 1- Why the positive pole is tested with extreme load while negative has not tested? The authors design DC/DC converter for both cases but only one case has been tested. 2- figures 17 and 18 show the results of current limited operation in short-circuit condition for positive firstly then bot poles but no figure shows when it start by negative then both? 3- No figure has been add to explain the main contribution of this paper, that mean, test for system with and without DC/DC converter and controller? 4-The authors did not discuss the cost issue of using DC/DC converter and controller compare to balance goals?

Author Response

Thanks for addressing English writing Problem. We did a complete proofreading of the paper by MDPI English editing services. Even after extensive English correction by them, we still repeatedly checked our manuscript to provide our paper with English correction as much as possible. 

1. Why the positive pole is tested with extreme load while negative has not tested? The authors design DC/DC converter for both cases but only one case has been tested.

Answer
  Thanks for the comment. In order to verify the extremely heavy asymmetric load condition of the negative pole, we have changed that Figure. 17(a) and added sentences in updated manuscript. We have added the following sentence in line 314-316.

(line 314-316)
In addition, the 3-level NPC AC/DC converter system also regulates the output voltage to 50V under the extremely heavy asymmetric load condition of the negative pole.

2. figures 17 and 18 show the results of current limited operation in short-circuit condition for positive firstly then bot poles but no figure shows when it start by negative then both?

Answer
  Thanks for the comment. We have added Figure. 18(a) and 18(b) to verify the current limit function in the short-circuit condition of the positive or negative pole respectively. In addition, we have added and improved the following sentence on lines 326-341 of updated manuscript.

(line 326-341)
Figure. 18 shows the current limit function of the proposed DC/DC converter in case of short-circuit condition. The short-circuit current was tested above 120A under experimental conditions. Figure. 18(a) and Figure. 18(b) verifies that the current limit function respectively operates at short-circuit condition in the positive or negative pole. When the short-circuit condition of 160 A occurs in the positive pole, the current of the positive pole is limited to 120A, and the voltage of the positive pole drops to the proper voltage to limit the 120A, as shown in Figure. 18(a). At this time, since the negative pole is not short-circuited, the load connected to the negative pole must be supplied with stable power. Therefore, the unbalanced output voltage caused by the short-circuit condition is controlled stably 750V at the negative pole. When the short-circuit condition of 135A occurs in the negative pole, the current of the negative pole is also limited to 120A, as shown in Figure. 18(b). Since the negative pole is not short-circuited, the output voltage of the positive pole is regulated by the DC/DC converter. Figure. 19 verifies the current limit function under the short-circuit condition of both poles. When the short-circuit condition of 175A occurs in the positive pole, the current limit function is operated to 120A. In addition, while the proposed DC/DC converter operates the current limit function at the positive pole, if the short-circuit condition also occurs at the negative pole, the current of the negative pole is also limited as 120A reference current as shown in Figure. 19.

3. No figure has been add to explain the main contribution of this paper, that mean, test for system with and without DC/DC converter and controller?

Answer
  Thanks for the comment. As comment by the reviewer, there is no experimental picture with DC/DC converter with and without. Figure. 15 and 16 (a) show that if the 3-level NPC AC/DC converter system composes without DC/DC converter, the unbalance of the output voltage occurs at extreme asymmetric loads. In the case of the short-circuit condition, if the 3-level NPC AC/DC converter system composes without DC/DC converter, the neutral point current is extremely generated, and the entire system is faulty and enters the protection operation.
  Since the prototype of the 300kW 3-level NPC AC/DC converter system is installed in the LVDC test bed in other regions, it is not possible to add a comparative test waveform at the present time. Therefore, we have added the following sentence on line 301-303 and line 324-326 of the updated manuscript to explain the need for the DC/DC converter.

(line 301-303)
If it is a conventional 3-level NPC AC/DC converter, the unbalance of the output voltage is consistently severe due to the limitation of the injected zero-sequence voltage method, entire system will fault.

(line 324-326)
In short-circuit condition, the excessive NP current flows to NP. The conventional 3-level NPC AC/DC converters can not control excessive NP current due to short-circuit condition, entire system will fault and stop.

4. The authors did not discuss the cost issue of using DC/DC converter and controller compare to balance goals?

Answer

Thanks for the comment. As comment by the reviewer, a cost issue arises when DC/DC converter and controller are added for balancing purposes only. However, converter for bipolar LVDC distribution must consider not only the balancing problem but also individual bidirectional operation and short-circuit conditions. In order to discuss the cost issue of the AC/DC converter system for bipolar LVDC distribution, we have added Figure. 10 and the following sentence on lines 189-199 of updated manuscript.

(line 189-199)
Figure. 10 shows that the AC/DC converter for the bipolar LVDC distribution can be composed of two 2-level AC / DC converters or a 3-level AC/DC converter. A 2-level AC/DC converter requires a three-winding transformer. Two 2-level AC/DC converters do not cause an unbalance problem in the bipolar output voltage. However, two additional DC/DC converters are eventually required to satisfy the above functions for bipolar LVDC distribution, as shown in Figure. 10(a).
The 3-level NPC AC/DC converter requires six more diodes than two 2-level AC/DC converters, but do not needed the three-winding transformer and reduces the required input filter. In addition, although the 3-level NPC AC/DC converters have the inherent problem of NP voltage unbalance, as shown in Figure. 10(b), one proposed DC/DC converter satisfies the essential functions for bipolar LVDC distribution, which has advantages in terms of price and equipment volume.

Reviewer 3 Report

Proofreading is a mandatory for the English grammars and style

Author Response

Proofreading is a mandatory for the English grammars and style

Answer

Thanks for addressing English writing Problem. We did a complete proofreading of the paper by MDPI English editing services. Even after extensive English correction by them, we still repeatedly checked our manuscript to provide our paper with English correction as much as possible.
